# Approximate Nearest Neighbor Negative Contrastive Learning for Dense Text Retrieval

**Lee Xiong,**[*] **Chenyan Xiong,**[*] **Ye Li, Kwok-Fung Tang, Jialin Liu,**
**Paul Bennett, Junaid Ahmed, Arnold Overwijk**
Microsoft Corporation.
`lexion, chenyan.xiong, yeli1, kwokfung.tang, jialliu,`
`paul.n.bennett, jahmed, arnold.overwijk@microsoft.com`

## Abstract

Conducting text retrieval in a learned dense representation space has many intriguing advantages. Yet dense retrieval (DR) often underperforms word-based sparse retrieval. In this paper, we first theoretically show the bottleneck of dense retrieval is the domination of uninformative negatives sampled in mini-batch training, which yield diminishing gradient norms, large gradient variances, and slow convergence. We then propose Approximate nearest neighbor Negative Contrastive Learning (ANCE), which selects hard training negatives globally from the entire corpus. Our experiments demonstrate the effectiveness of ANCE on web search, question answering, and in a commercial search engine, showing ANCE dot-product retrieval nearly matches the accuracy of BERT-based cascade IR pipeline. We also empirically validate our theory that negative sampling with ANCE better approximates the oracle importance sampling procedure and improves learning convergence.

## 1 Introduction

Many language systems rely on text retrieval as their first step to find relevant information. For example, search ranking (Nogueira & Cho, 2019), open domain question answering (OpenQA) (Chen et al., 2017), and fact verification (Thorne et al., 2018) all first retrieve relevant documents for their later stage reranking, machine reading, and reasoning models. All these later-stage models enjoy the advancements of deep learning techniques (Rajpurkar et al., 2016; Wang et al., 2019), while, the first stage retrieval still mainly relies on matching discrete bag-of-words, e.g., BM25, which has become the pain point of many systems (Nogueira & Cho, 2019; Luan et al., 2020; Zhao et al., 2020).

Dense Retrieval (DR) aims to overcome the sparse retrieval bottleneck by matching in a continuous representation space learned via neural networks (Lee et al., 2019; Karpukhin et al., 2020; Luan et al., 2020). It has many desired properties: fully learnable representation, easy integration with pretraining, and efficiency support from approximate nearest neighbor (ANN) search (Johnson et al., 2017). These grant dense retrieval an intriguing potential to fundamentally overcome some intrinsic limitations of sparse retrieval, for example, vocabulary mismatch (Croft et al., 2009).

One challenge in dense retrieval is to construct proper negative instances when learning the representation space (Karpukhin et al., 2020). Unlike in reranking (Liu, 2009) where the training and testing negatives are both irrelevant documents from previous retrieval stages, in first stage retrieval, DR models need to distinguish *all irrelevant ones* in a corpus with millions or billions of documents. As illustrated in Fig. 1, these negatives are quite different from those retrieved by sparse models.

Recent research explored various ways to construct negative training instances for dense retrieval (Karpukhin et al., 2020), e.g., using contrastive learning (Oord et al., 2018; He et al., 2020; Chen et al., 2020a) to select hard negatives in current or recent mini-batches. However, as observed in recent research (Karpukhin et al., 2020), the in-batch local negatives, though effective in learning word or visual representations, are not significantly better than spare-retrieved negatives in representation learning for dense retrieval. In addition, the accuracy of dense retrieval models often underperform BM25, especially on documents (Gao et al., 2020b; Luan et al., 2020).

---

[*]Lee and Chenyan contributed equally.

In this paper, we first theoretically analyze the convergence of dense retrieval training with negative sampling. Using the variance reduction framework (Alain et al., 2015; Katharopoulos & Fleuret, 2018), we show that, under conditions commonly met in dense retrieval, local in-batch negatives lead to diminishing gradient norms, resulted in high stochastic gradient variances and slow training convergence — the local negative sampling is the bottleneck of dense retrieval's effectiveness.

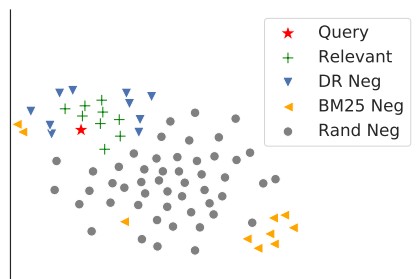

Based on our analysis, we propose Approximate nearest neighbor Negative Contrastive Estimation (ANCE), a new contrastive representation learning mechanism for dense retrieval. Instead of random or in-batch local negatives, ANCE constructs global negatives using the being-optimized DR model to retrieve from the entire corpus. This fundamentally aligns the distribution of negative samples in training and of irrelevant documents to separate in testing. From the variance reduction perspective, these ANCE negatives lift the upper bound of per instance gradient norm, reduce the variance of the stochastic gradient estimation, and lead to faster learning convergence.

Figure 1: T-SNE (Maaten & Hinton, 2008) representations of query, relevant documents, negative training instances from BM25 (BM25 Neg) or randomly sampled (Rand Neg), and testing negatives (DR Neg) in dense retrieval.

We implement ANCE using an asynchronously updated ANN index of the corpus representation. Similar to Guu et al. (2020), we maintain an Inferencer that parallelly computes the document encodings with a recent checkpoint from the being optimized DR model, and refresh the ANN index used for negative sampling once it finishes, to keep up with the model training. Our experiments demonstrate the advantage of ANCE in three text retrieval scenarios: standard web search (Craswell et al., 2020), OpenQA (Rajpurkar et al., 2016; Kwiatkowski et al., 2019), and in a commercial search engine's retrieval system. We also empirically validate our theory that the gradient norms on ANCE sampled negatives are much bigger than local negatives, thus improving the convergence of dense retrieval models.[1]

## 2 PRELIMINARIES

In this section, we discuss the preliminaries of dense retrieval and its representation learning.

**Task Definition:** Given a query $q$ and a corpus $C$, the first stage retrieval is to find a set of documents relevant to the query $D^+ = \{d_1, ..., d_i, ..., d_n\}$ from $C$ ($|D^+| \ll |C|$), which then serve as input to later more complex models (Croft et al., 2009). Instead of using sparse term matches and inverted index, *Dense Retrieval* calculates the retrieval score $f()$ using similarities in a learned embedding space (Lee et al., 2019; Luan et al., 2020; Karpukhin et al., 2020):

$$f(q, d) = \text{sim}(g(q; \theta), g(d; \theta)), \tag{1}$$

where $g()$ is the representation model that encodes the query or document to dense embeddings. The encoder parameter $\theta$ provides the main capacity. The similarity function (sim()) is often simply cosine or dot product to leverage efficient ANN retrieval (Johnson et al., 2017; Guo et al., 2020).

**BERT-Siamese Model:** A standard instantiation of Eqn. 1 is to use the BERT-Siamese/two-tower/dual-encoder model (Lee et al., 2019; Karpukhin et al., 2020; Luan et al., 2020):

$$f(q, d) = \text{BERT}(q) \cdot \text{BERT}(d) = \text{MLP}([\vec{\text{CLS}}]_q) \cdot \text{MLP}([\vec{\text{CLS}}]_d). \tag{2}$$

It encodes the query and document separately with BERT as the encoder $g()$, using their last layer's [CLS] token representation, and applied dot product ($\cdot$) on them. This enables offline pre-computing of the document encodings and efficient first-stage retrieval. In comparison, the BERT reranker (Nogueira et al., 2019) applies BERT on the concatenation of each to-rerank query-document pair: $\text{BERT}(q \circ d)$, which has explicit access to term level interactions between query-document with transformer attentions, but is often infeasible in first stage retrieval as enumerating all documents in the corpus for each query is too costly.

---

[1]Our code and trained models are available at http://aka.ms/ance.

**Learning with Negative Sampling:** The effectiveness of DR resides in learning a good representation space that maps query and relevant documents together, while separating irrelevant ones. The learning of this representation often follows standard learning to rank (Liu, 2009): Given a query $q$, a set of its relevant document $D_q^+$ and irrelevant ones $D_q^-$, find the best $\theta^*$ that:

$$\theta^* = \text{argmin}_\theta \sum_q \sum_{d^+ \in D_q^+} \sum_{d^- \in D_q^-} l(f(q, d^+), f(q, d^-)). \tag{3}$$

The loss $l()$ can be binary cross entropy (BCE), hinge loss, or negative log likelihood (NLL).

A unique challenge in dense retrieval, targeting first stage retrieval, is that the irrelevant documents to separate are from the entire corpus ($D_q^- = C \setminus D_q^+$). This often leads to millions of negative instances, which have to be sampled in training:

$$\theta^* = \text{argmin}_\theta \sum_q \sum_{d^+ \in D^+} \sum_{d^- \in \hat{D}^-} l(f(q, d^+), f(q, d^-)). \tag{4}$$

Here we start to omit the subscript $q$ in $D_q$. All $D^+$ and $D^-$ are query dependent. A natural choice is to sample negatives $\hat{D}^-$ from top documents retrieved by BM25. However, they may bias the DR model to merely mimic sparse retrieval (Luan et al., 2020). Another way is to sample negatives in local mini-batches, e.g., as in contrastive learning (Oord et al., 2018), however, these local negatives do not significantly outperform BM25 negatives (Karpukhin et al., 2020; Luan et al., 2020).

## 3 Analyses on The Convergence of Dense Retrieval Training

In this section, we theoretically analyze the convergence of dense retrieval training. We first show the connections between learning convergence and gradient norms (Sec. 3.1), then we discuss how non-informative negatives in dense retrieval yield less optimal convergence (Sec. 3.2).

### 3.1 Oracle Negative Sampling According to Per-Instance Gradient-Norm

Let $l(d^+, d^-) = l(f(q, d^+), f(q, d^-)$ be the loss function on the training triple $(q, d^+, d^-)$, $P_{D^-}$ the negative sampling distribution for the given $(q, d^+)$, and $p_{d^-}$ the sampling probability of negative instance $d^-$, a stochastic gradient decent (SGD) step with importance sampling (Alain et al., 2015) is:

$$\theta_{t+1} = \theta_t - \eta \frac{1}{Np_{d^-}} \nabla_{\theta_t} l(d^+, d^-), \tag{5}$$

with $\theta_t$ the parameter at $t$-th step, $\theta_{t+1}$ the one after, and $N$ the total number of negatives. The scaling factor $\frac{1}{Np_{d^-}}$ ensures Eqn. 5 is an unbiased estimator of the non-stochastic gradient on the full data.

Then we can characterize the converge rate of this SGD step as the movement to the optimal $\theta^*$. Following derivations in variance reduction (Katharopoulos & Fleuret, 2018; Johnson & Guestrin, 2018), let $g_{d^-} = \frac{1}{Np_{d^-}} \nabla_{\theta_t} l(d^+, d^-)$ the weighted gradient, the convergence rate is:

$$\mathbb{E}\Delta^t = ||\theta_t - \theta^*||^2 - \mathbb{E}_{P_{D^-}}(||\theta_{t+1} - \theta^*||^2) \tag{6}$$

$$= ||\theta_t||^2 - 2\theta_t^T \theta^* - \mathbb{E}_{P_{D^-}}(||\theta_t - \eta g_{d^-}||^2) + 2\theta^{*T} \mathbb{E}_{P_{D^-}}(\theta_t - \eta g_{d^-}) \tag{7}$$

$$= -\eta^2 \mathbb{E}_{P_{D^-}}(||g_{d^-}||^2) + 2\eta \theta_t^T \mathbb{E}_{P_{D^-}}(g_{d^-}) - 2\eta \theta^{*T} \mathbb{E}_{P_{D^-}}(g_{d^-}) \tag{8}$$

$$= 2\eta \mathbb{E}_{P_{D^-}}(g_{d^-})^T(\theta_t - \theta^*) - \eta^2 \mathbb{E}_{P_{D^-}}(||g_{d^-}||^2) \tag{9}$$

$$= 2\eta \mathbb{E}_{P_{D^-}}(g_{d^-})^T(\theta_t - \theta^*) - \eta^2 \mathbb{E}_{P_{D^-}}(g_{d^-})^T \mathbb{E}_{P_{D^-}}(g_{d^-}) - \eta^2 \text{Tr}(\mathcal{V}_{P_{D^-}}(g_{d^-})). \tag{10}$$

This shows we can obtain better convergence rate by sampling from a distribution $P_{D^-}$ that minimizes the variance of the *stochastic gradient estimator* $\mathbb{E}_{P_{D^-}}(||g_{d^-}||^2)$, or $\text{Tr}(\mathcal{V}_{P_{D^-}}(g_{d^-}))$ as the estimator is unbiased. The variance reflects how good the stochastic gradient from negative sampling represents the full gradient on all negatives—the latter is ideal but infeasible. Intuitively, we prefer the stochastic estimator to be stable and have smaller variances.

A well known result in importance sampling (Alain et al., 2015; Johnson & Guestrin, 2018) is that there exists an optimal distribution that:

$$p_{d^-}^* = \text{argmin}_{p_{d^-}} \text{Tr}(\mathcal{V}_{P_{D^-}}(g_{d^-})) \propto ||\nabla_{\theta_t} l(d^+, d^-)||_2. \tag{11}$$

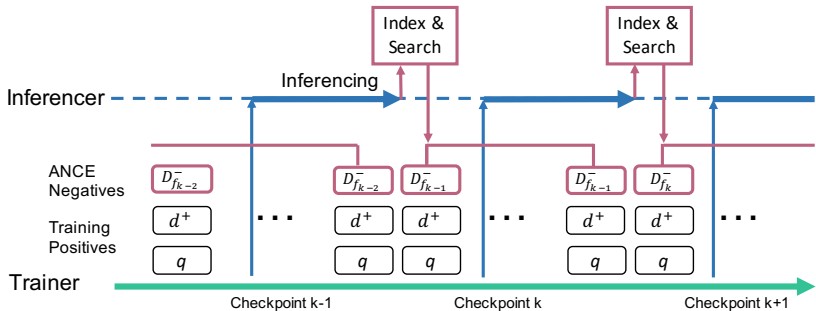

Figure 2: ANCE Asynchronous Training. The Trainer learns the representation using negatives from the ANN index. The Inferencer uses a recent checkpoint to update the representation of documents in the corpus and, once finished, refreshes the ANN index with most up-to-date encodings.

To prove this, one can apply Jensen's inequality on the gradient variance and verify that Eqn. 11 achieves the minimum. The detailed derivations can be find in Johnson & Guestrin (2018).

Eqn. 11 shows that the convergence rate can be improved by sampling negatives proportional to their per-instance gradient norms (though too expensive to calculate). Intuitively, an negative instance with larger gradient norm is more likely to reduce the non-stochastic training loss, thus should be sampled more frequently than those with diminishing gradients. The correlation of larger gradient norm and better training convergence is also observed in BERT fine-tuning (Mosbach et al., 2021).

### 3.2 Uninformative In-Batch Negatives and Their Diminishing Gradients

**Diminishing Gradients of Uninformative Negatives:** Though the close form of gradient norms often does not exist, Katharopoulos & Fleuret (2018) derives the following upper bound:

$$||\nabla_{\theta_t} l(d^+, d^-)||_2 \leq L\rho ||\nabla_{\phi_L} l(d^+, d^-)||_2, \tag{12}$$

where L is the number of layers, $\rho$ is composed by pre-activation weights and gradients in intermediate layers, and $||\nabla_{\phi_L} l(d^+, d^-)||_2$ is the gradient on the last layer. The derivation of this upper bound is on multi-layer perception with any depths and any activation function that is Lipschitz continuous (Katharopoulos & Fleuret, 2018). On complicated neural networks, the intermediate layers are regulated by various normalization and this upper bound often holds empirically (Sec. 6.3).

In addition, for many loss functions, for example, BCE loss and pairwise hinge loss, we can verify that when the loss goes to zero the gradient norm of the last layer also goes to zero: $l(d^+, d^-) \to 0 \Rightarrow ||\nabla_{\phi_L} l(d^+, d^-)||_2 \to 0$ (Katharopoulos & Fleuret, 2018).

Putting everything together, using uninformative negative samples with near zero loss results in the following chain of undesirable properties:

$$\underbrace{||\nabla_{\phi_L} l(d^+, d^-)||_2 \to 0}_{\text{low upper bound}} \Rightarrow \underbrace{||\nabla_{\theta_t} l(d^+, d^-)||_2 \to 0}_{\text{diminishing gradient norm}} \Rightarrow \underbrace{\text{Tr}(\mathcal{V}_{P_{D^-}}(g_{d^-})) \uparrow}_{\text{large scholastic variance}} \Rightarrow \underbrace{\mathbb{E}\Delta^t \downarrow}_{\text{slow convergence}}. \tag{13}$$

The uninformative negative samples yield diminishing gradient norms, larger variances of the scholastic gradient estimator, and less optimal learning convergence.

**Inefficacy of Local In-Batch Negatives:** We argue that, when training DR models, the in-batch local negatives are unlikely to provide informative samples due to two properties of text retrieval.

Let $D^{-*}$ be the set of informative negatives that are hard to distinguish from $D^+$, and $b$ be the batch size, we have (1) $b \ll |C|$, the batch size is far smaller than the corpus size; (2) $|D^{-*}| \ll |C|$, that only a few negatives are informative and the majority of corpus is trivially unrelated.

Both conditions hold in most dense retrieval scenarios. The two together make the probability that a random mini-batch includes meaningful negatives ($p = \frac{b|D^{-*}|}{|C|^2}$) close to zero. Selecting negatives from local training batches unlikely provides optimal training signals for dense retrieval.

## 4  APPROXIMATE NEAREST NEIGHBOR NOISE CONTRASTIVE ESTIMATION

Our analyses show the importance, if not necessity, to construct negatives *globally* from the corpus to avoid uninformative negatives for better learning convergence. In this section, we propose *Approximate nearest neighbor Negative Contrastive Estimation* (ANCE), which selects negatives from the entire corpus using an asynchronously updated ANN index.

**ANCE** samples negatives from the top retrieved documents via the DR model from the ANN index:

$$\theta^* = \mathrm{argmin}_\theta \sum_q \sum_{d^+ \in D^+} \sum_{d^- \in D^-_{\mathrm{ANCE}}} l(f(q, d^+), f(q, d^-)),\qquad(14)$$

with $D^-_{\mathrm{ANCE}} = \mathrm{ANN}_{f(q,d)} \setminus D^+$ and $\mathrm{ANN}_{f(q,d)}$ the top retrieved documents by $f()$ from the ANN index. By definition, $D^-_{\mathrm{ANCE}}$ are the hardest negatives for the current DR model: $D^-_{\mathrm{ANCE}} \approx D^{-*}$. In theory, these more informative negatives have higher training loss, elevate the upper bound on the gradient norms (first component of Eqn 13), and prevent the slow convergence indicated in Eqn 13.

ANCE can pair with many DR models. For simplicity, we use BERT-Siamese (Eqn. 2), with shared encoder weights between $q$ and $d$ and negative log likelihood (NLL) loss (Luan et al., 2020).

**Asynchronous Index Refresh:** During stochastic training, the DR model $f()$ is updated each mini-batch. Maintaining an update-to-date ANN index to select fresh ANCE negatives is challenging, as the index update requires two operations: 1) *Inference*: refresh the representations of all documents in the corpus with an updated DR model; 2) *Index*: rebuild the ANN index using updated representations. Although *Index* is efficient (Johnson et al., 2017), *Inference* is too expensive to compute per batch as it requires a forward pass on a corpus much bigger than a training batch.

Thus we implement an asynchronous index refresh similar to Guu et al. (2020), and update the ANN index once every $m$ batches, i.e., with checkpoint $f_k$. As illustrated in Fig. 2, besides the Trainer, we run an Inferencer that takes the latest checkpoint (e.g., $f_k$) and recomputes the encodings of the entire corpus. In parallel, the Trainer continues its stochastic learning using $D^-_{f_{k-1}}$ from $\mathrm{ANN}_{f_{k-1}}$. Once the corpus is re-encoded, the Inferencer updates the index ($\mathrm{ANN}_{f_k}$) and feed it to the Trainer, e.g., through a shared file system. In this process, the ANCE negatives ($D^-_{\mathrm{ANCE}}$) are asynchronously updated to "catch up" with the stochastic training, with an async-gap determined by the computing resources allocated to the Inferencer. Our experiment in Sec 6.4 studies the influence of this async-gap in learning convergence.

## 5  EXPERIMENTAL METHODOLOGIES

This section describes our experimental setups. More details can be found in Appendix A.1 and A.2.

**Benchmarks:** The web search experiments use the TREC 2019 Deep Learning (DL) Track (Craswell et al., 2020). It is a standard ad hoc retrieval benchmark: given web queries from Bing, to retrieval passages or documents from the MS MARCO corpus (Bajaj et al., 2016). We use the official setting and focus on the first stage retrieval, but also show results when reranking top 100 BM25 candidates.

The OpenQA experiments use the Natural Questions (NQ) (Kwiatkowski et al., 2019) and TriviaQA (TQA) (Joshi et al., 2017), following the exact settings from Karpukhin et al. (2020). The metrics are Coverage@20/100, which evaluate whether the Top-20/100 retrieved passages include the answer. We also evaluate whether ANCE's better retrieval can propagate to better answer accuracy, by running the state-of-the-art systems' readers on top of ANCE retrieval. The readers are RAG-Token (Lewis et al., 2020b) on NQ and DPR Reader on TQA, using their suggested settings.

We also study the effectiveness of ANCE in the first stage retrieval of a commercial search engine's production system. We change the training of a production-quality DR model to ANCE, and evaluate the offline gains in various corpus sizes, encoding dimensions, and exact/approximate search.

**Baselines:** In TREC DL, we include best runs in relevant categories and refer to Craswell et al. (2020) for more baseline scores. We implement various DR baselines using the same BERT-Siamese (Eqn. 2), but with different training negative construction: random sampling in batch (Rand Neg), random sampling from BM25 top 100 (BM25 Neg) (Lee et al., 2019; Gao et al., 2020b), and the 1:1 combination of BM25 and Random negatives (BM25 + Rand Neg) (Karpukhin et al., 2020; Luan

Table 1: Results in TREC 2019 Deep Learning Track. Results not available are marked as "n.a.", not applicable are marked as "–". Best results in each category are marked bold. Dense Retrieval baselines use the same BERT-Siamese but different training strategies.

| | MARCO Dev Passage Retrieval | | TREC DL Passage NDCG@10 | | TREC DL Document NDCG@10 | |
|---|---|---|---|---|---|---|
| | MRR@10 | Recall@1k | Rerank | Retrieval | Rerank | Retrieval |
| **Sparse & Cascade IR** | | | | | | |
| BM25 | 0.240 | 0.814 | – | 0.506 | – | 0.519 |
| Best DeepCT | 0.243 | n.a. | – | n.a. | – | 0.554 |
| Best TREC Trad Retrieval | 0.240 | n.a. | – | 0.554 | – | 0.549 |
| BERT Reranker | – | – | **0.742** | – | 0.646 | – |
| **Dense Retrieval** | | | | | | |
| Rand Neg | 0.261 | 0.949 | 0.605 | 0.552 | 0.615 | 0.543 |
| NCE Neg | 0.256 | 0.943 | 0.602 | 0.539 | 0.618 | 0.542 |
| BM25 Neg | 0.299 | 0.928 | 0.664 | 0.591 | 0.626 | 0.529 |
| DPR (BM25 + Rand Neg) | 0.311 | 0.952 | 0.653 | 0.600 | 0.629 | 0.557 |
| BM25 $\rightarrow$ Rand | 0.280 | 0.948 | 0.609 | 0.576 | 0.637 | 0.566 |
| BM25 $\rightarrow$ NCE Neg | 0.279 | 0.942 | 0.608 | 0.571 | 0.638 | 0.564 |
| BM25 $\rightarrow$ BM25 + Rand | 0.306 | 0.939 | 0.648 | 0.591 | 0.626 | 0.540 |
| ANCE (FirstP) | **0.330** | **0.959** | 0.677 | **0.648** | 0.641 | 0.615 |
| ANCE (MaxP) | – | – | – | – | **0.671** | **0.628** |

Table 2: Retrieval results (Answer Coverage at Top-20/100) on Natural Questions (NQ) and Trivial QA (TQA) in the setting from Karpukhin et al. (2020).

| | Single Task | | Multi Task | |
|---|---|---|---|---|
| | NQ | TQA | NQ | TQA |
| Retriever | Top-20/100 | Top-20/100 | Top-20/100 | Top-20/100 |
| BM25 | 59.1/73.7 | 66.9/76.7 | –/– | –/– |
| DPR | 78.4/85.4 | 79.4/85.0 | 79.4/86.0 | 78.8/84.7 |
| BM25+DPR | 76.6/83.8 | 79.8/84.5 | 78.0/83.9 | 79.9/84.4 |
| ANCE | **81.9/87.5** | **80.3/85.3** | **82.1/87.9** | **80.3/85.2** |

Table 3: Relative gains in the first stage retrieval of a commercial search engine. The gains are from changing the training of a production DR model to ANCE.

| Corpus Size | Dim | Search | Gain |
|---|---|---|---|
| 250 Million | 768 | KNN | +18.4% |
| 8 Billion | 64 | KNN | +14.2% |
| 8 Billion | 64 | ANN | +15.5% |

et al., 2020). We also compare with contrastive learning/Noise Contrastive Estimation, which uses hardest negatives in batch (NCE Neg) (Gutmann & Hyvärinen, 2010; Oord et al., 2018; Chen et al., 2020a). In OpenQA, we compare with DPR, BM25, and their combinations (Karpukhin et al., 2020).

**Implementation Details:** In TREC DL, recent research found MARCO passage training labels cleaner (Yan et al., 2019) and BM25 negatives can help train dense retrieval (Karpukhin et al., 2020; Luan et al., 2020). Thus, we include a "BM25 Warm Up" setting (BM25 $\rightarrow$ *), where the DR models are first trained using MARCO official BM25 Negatives. ANCE is also warmed up by BM25 negatives. All DR models in TREC DL are fine-tuned from RoBERTa base uncased (Liu et al., 2019). In OpenQA, we warm up ANCE using the released DPR checkpoints (Karpukhin et al., 2020).

To fit long documents in BERT-Siamese, ANCE uses the two settings from Dai & Callan (2019b), FirstP which uses the first 512 tokens of the document, and MaxP, where the document is split to 512-token passages (maximum 4) and the passage level scores are max-pooled. The max-pooling operation is natively supported in ANN. The ANN search uses the Faiss IndexFlatIP Index (Johnson et al., 2017). We use batch size 8 and gradient accumulation step 2 on 4 V100 32GB GPUs. For each positive, we uniformly sample one negative from ANN top 200 (weighted sample and/or from top 100 also work well). We measured ANCE efficiency using one 32GB V100 GPU, Intel(R) Xeon(R) Platinum 8168 CPU and 650GB of RAM memory.

In asynchronous training, we allocate equal amounts of GPUs to the Trainer and the Inferencer, often four or eight each. The Trainer produces a model checkpoint every 5k or 10k training batches. The Inferencer loads the recent model checkpoint and calculates the embeddings of the corpus in parallel. Once the embedding calculation finishes, a new ANN index is built and the Trainer switches to it for negative construction. All their communications are through a shared file system. On MS MARCO, the ANN negative index is refreshed about every 10K training steps.

Table 4: OpenQA Test Scores in Single Task Setting. ANCE+Reader switches the retrieve of the OpenQA systems from DPR to ANCE and keeps their QA components, which is RAG-Token on Natural Questions (NQ) and DPR Reader on Trivia QA (TQA). T5 results are "closed-book". The others are open-book.

| Model | NQ | TQA |
|---|---|---|
| T5-11B (Closed) (Roberts et al., 2020) | 34.5 | - |
| T5-11B + SSM (Closed) (Roberts et al., 2020) | 36.6 | - |
| REALM (Guu et al., 2020) | 40.4 | - |
| DPR (Karpukhin et al., 2020) | 41.5 | 56.8 |
| DPR + BM25 (Karpukhin et al., 2020) | 39.0 | 57.0 |
| RAG-Token (Lewis et al., 2020b) | 44.1 | 55.2 |
| RAG-Sequence (Lewis et al., 2020b) | 44.5 | 56.1 |
| ANCE + Reader | **46.0** | **57.5** |

Table 5: Efficiency of ANCE Serving and Training.

| Operation | Offline | Online |
|---|---|---|
| BM25 Index Build | 3h | – |
| BM25 Retrieval | – | 37ms |
| BERT Rerank | – | 1.15s |
| Sparse IR Total (BM25 + BERT) | – | **1.42s** |
| **ANCE Inference** | | |
| Encoding of Corpus/Per doc | 10h/4.5ms | – |
| Query Encoding | – | 2.6ms |
| ANN Retrieval (batched q) | – | 9ms |
| Dense Retrieval Total | – | **11.6ms** |
| **ANCE Training** | | |
| Encoding of Corpus/Per doc | 10h/4.5ms | – |
| ANN Index Build | 10s | – |
| Neg Construction Per Batch | 72ms | – |
| Back Propagation Per Batch | 19ms | – |

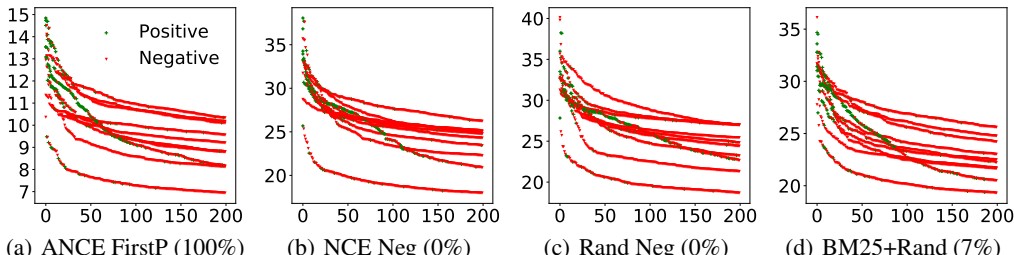

(a) ANCE FirstP (100%)   (b) NCE Neg (0%)   (c) Rand Neg (0%)   (d) BM25+Rand (7%)

Figure 3: The top DR scores for ten random TREC DL testing queries. The x-axes are their ranking order. The y-axes are their retrieval scores minus corpus average. All models are warmed up by BM25 Neg. The percentages are the overlaps between the testing and training negatives near convergence.

## 6 EVALUATION RESULTS

In this section, we first evaluate the effectiveness and efficiency of ANCE. Then we empirically study the convergence of ANCE training and the influence of the asynchronous gap. More comparisons of dense and sparse retrieval, hyperparameter study, and case study are in Appendix.

### 6.1 EFFECTIVENESS

In web search (Table 1), ANCE significantly outperforms all sparse retrieval, including the BERT-based DeepCT (Dai et al., 2019). Among DR models with different training strategies, ANCE is the only one robustly exceeding sparse methods in document retrieval. In OpenQA, ANCE outperforms DPR and its fusion with BM25 (DPR+BM25) in retrieval accuracy (Table 2). It also improves end-to-end QA accuracy, using the same readers with previous state-of-the-arts but ANCE retriever (Table 4). ANCE's effectiveness is even more observed in real production (Table 3).

Among all DR models, ANCE has the smallest gap between its retrieval and reranking accuracy, showing the importance of global negatives in training retrieval models. ANCE retrieval nearly matches the accuracy of the cascade IR with interaction-based BERT Reranker (Nogueira & Cho, 2019), even though BERT-Siamese does not explicitly capture term-level interactions. *With ANCE, we can learn a representation space that effectively captures the finesse of search relevance.*

### 6.2 EFFICIENCY

The efficiency of ANCE (FirstP) in TREC DL doc is shown in Table 5. In *serving*, we measure the online latency to retrieve/rank 100 documents per query, with query batched. DR is 100x faster than BERT Rerank, a natural benefit of BERT-Siamese where the document encodings are calculated offline and separately with the query. In comparison, the interaction-based BERT Reranker runs BERT once per query and candidate document pair. The bulk of training computing is in calculating the encoding of the corpus for ANCE negative construction, which is mitigated by making the index refresh asynchronous.

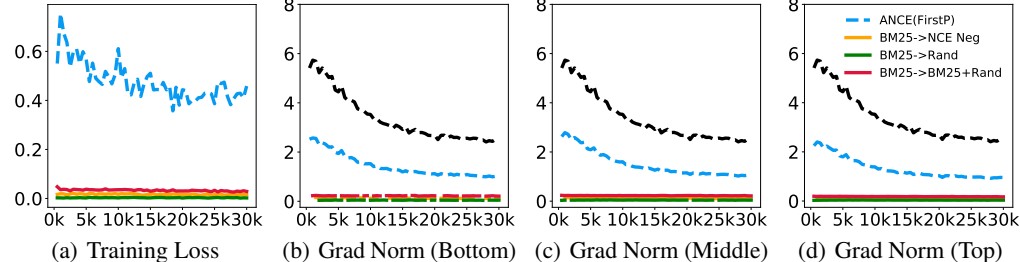

(a) Training Loss    (b) Grad Norm (Bottom)    (c) Grad Norm (Middle)    (d) Grad Norm (Top)

Figure 4: The loss and gradient norms during DR training (after BM25 warm up). The gradient norms are the per-layer average of the bottom (1-4), middle (5-8), and top (9-12) transformer layers. Black dotted lines are the grad norm of the last layer in ANCE (FirstP). The x-axes are training steps.

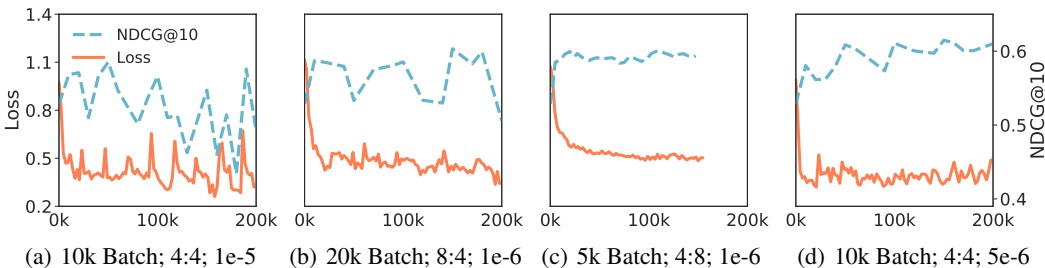

(a) 10k Batch; 4:4; 1e-5    (b) 20k Batch; 8:4; 1e-6    (c) 5k Batch; 4:8; 1e-6    (d) 10k Batch; 4:4; 5e-6

Figure 5: Training loss and testing NDCG of ANCE (FirstP) on documents. The sub captions list the ANN index refreshing rate (e.g., per 10k Batch), Trainer:Inferencer GPU allocation (e.g., 4:4), and learning rate (e.g., 1e-5). The x-axes are the training steps.

## 6.3 EMPIRICAL ANALYSES ON TRAINING CONVERGENCE

We first show the long tail distribution of search relevance in dense retrieval. As plotted in Fig. 3, there are a few instances per query with significant higher retrieval scores, while the majority form a long tail. In retrieval/ranking, the key challenge is to distinguish the relevant ones among those highest scored ones; the rest is trivially irrelevant. We also empirically measure the probability of local in-batch negatives including informative negatives ($D^{-*}$), by their overlap with top 100 highest scored negatives. This probability, either using NCE Neg or Rand Neg, is *zero*, the same as our theory shows. In comparison, the overlap between BM25 Neg with top dense retrieved negatives is 15%, while that of ANCE negatives starts at 63% and converges to 100% by design.

Then we empirically validate our theory that local negatives lead to lower loss, bounded gradient norm, non-ideal importance sampling, and thus slow convergence (Eqn. 13). The training loss and pre-clip gradient norms during DR training are plotted in Fig. 4. As expected, the uninformative local negatives resulted in near-zero gradient norms, while ANCE global negatives maintain a higher gradient norm. The gradient norm of the last layer in the BERT-Siamese model during ANCE training (black dotted lines in Fig. 4) is consistently bigger than the other layers, which empirically aligns with the upper bound in Eqn. 12. Also as our theory suggests, the gradient norms of local negatives are bounded close to zero, while those of ANCE global negatives are bigger by orders of magnitude. This confirms that ANCE better approximates the oracle importance sampling distribution ($p^*_{d^-} \propto ||\nabla_{\theta_t} l(d^+, d^-)||_2$) and improves learning convergence.

## 6.4 IMPACT OF ASYNCHRONOUS GAP

The efficiency constraints enforce an asynchronous gap (async-gap) in ANCE training: The negatives are selected using the encodings from an earlier stage of the being optimized DR model. The async-gap is determined by the target index refreshing rate, which is determined by the allocation of computing resource on the Trainer versus the Inferencer, as well as the learning rate. This experiment studies the impact of this async-gap. The training curves and testing NDCG of different configurations are plotted in Fig. 5.

A too large async-gap, either from large learning rate (Fig. 5(a)) or low refreshing rate (Fig. 5(b)), makes the training unstable, perhaps because the refreshed index changes too dramatically, as indicated by the peaks in training loss and dips of testing NDCG. The async-gap is not significant when we allocate an equal amount of GPUs to the index refreshing and to the training (Fig. 5(d)). Further reducing the gap (Fig. 5(c)) does not improve learning convergence.

In many scenarios, using a same amount of extra GPUs for ANCE as a one-time training cost is a good return of investment. The efficiency bottleneck in production is often in inference and serving.

## 7 RELATED WORK

In early research on neural information retrieval (Neu-IR) (Mitra & Craswell, 2018), a common belief was that the interaction models, those that explicitly handle term level matches, are more effective though more expensive (Guo et al., 2016; Xiong et al., 2017; Nogueira & Cho, 2019). Many techniques are developed to reduce their cost, for example, distillation (Gao et al., 2020a) and caching (Humeau et al., 2020; Khattab & Zaharia, 2020; MacAvaney et al., 2020). ANCE shows that a properly trained representation-based BERT-Siamese can be effective as well. This finding will motivate many new research explorations in Neu-IR.

Deep learning has been used to improve various components of sparse retrieval, for example, term weighting (Dai & Callan, 2019b), query expansion (Zheng et al., 2020), and document expansion (Nogueira et al., 2019). Dense Retrieval chooses a different path and conducts retrieval purely in the embedding space via ANN search (Lee et al., 2019; Chang et al., 2020; Karpukhin et al., 2020; Luan et al., 2020). This work demonstrates that a simple dense retrieval system can achieve SOTA accuracy, while also behaves dramatically different from sparse retrieval. The recent advancement in dense retrieval may raise a new generation of search systems.

Recent research in contrastive representation learning also shows the benefits of sampling negatives from a larger candidate pool. In computer vision, He et al. (2020) decouple the negative sampling pool size with training batch size, by maintaining a negative candidate pool of recent batches and updating their representation with momentum. This enlarged negative pool significantly improves unsupervised visual representation learning (Chen et al., 2020b). A parallel work (Xiong et al., 2021) improves DPR by sampling negatives from a memory bank (Wu et al., 2018) — in which the representations of negative candidates are frozen so more candidates can be stored. Instead of a bigger local pool, ANCE goes all the way along this trajectory and constructs negatives globally from the entire corpus, using an asynchronously updated ANN index.

Besides being a real world application itself, dense retrieval is also a core component in many other language systems, for example, to retrieve relevant information for grounded language models (Khandelwal et al., 2020; Guu et al., 2020), extractive/generative QA (Karpukhin et al., 2020; Lewis et al., 2020b), and fact verification (Xiong et al., 2021), or to find paraphrase pairs for pretraining (Lewis et al., 2020a). There dense retrieval models are either frozen or optimized indirectly by signals from their end tasks. ANCE is orthogonal to those lines of research and focuses on the representation learning for dense retrieval. Its better retrieval accuracy can benefit many other language systems.

## 8 CONCLUSION

In this paper, we first provide theoretical analyses on the convergence of representation learning in dense retrieval. We show that under common conditions in text retrieval, the local negatives used in DR training are uninformative, yield low gradient norms, and contribute little to the learning convergence. We then propose ANCE to eliminate this bottleneck by constructing training negatives globally from the entire corpus. Our experiments demonstrate the advantage of ANCE in web search, OpenQA, and the production environment of a commercial search engine. Our studies empirically validate our theory that ANCE negatives have much bigger gradient norms, reduce the stochastic gradient variance, and improve training convergence.

## 9 ACKNOWLEDGMENTS

We thank Di He for discussions on learning theories and Safoora Yousefi for feedback in writing.

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

# A APPENDIX

## A.1 MORE EXPERIMENTAL DETAILS

**More Details on TREC DL Benchmarks:** There are two tasks in the TREC DL 2019 Track: document retrieval and passage retrieval. The training and development sets are from MS MARCO, which includes passage level relevance labels for one million Bing queries (Bajaj et al., 2016). The document corpus was post-constructed by back-filling the body texts of the passage's URLs and their labels were inherited from its passages (Craswell et al., 2020). The testing sets are labeled by NIST accessors on the top 10 ranked results from past Track participants (Craswell et al., 2020).

TREC DL official metrics include NDCG@10 on test and MRR@10 on MARCO Passage Dev. MARCO Document Dev is noisy and the recall on the DL Track testing is less meaningful due to low label coverage on DR results. There is a two-year gap between the construction of the passage training data and the back-filling of their full document content. Some original documents were no longer available. There is also a decent amount of content changes in those documents during the two-year gap, and many no longer contain the passages. This back-filling perhaps is the reason why many Track participants found the passage training data is more effective than the inherited document labels. Note that the TREC testing labels are not influenced as the annotators were provided the same document contents when judging.

All the TREC DL runs are trained using these training data. Their inference results on the testing queries of the document and the passage retrieval tasks were evaluated by NIST assessors in the standard TREC-style pooling technique (Voorhees, 2000). The pooling depth is set to 10, that is, the top 10 ranked results from all participated runs are evaluated, and these evaluated labels are released as the official TREC DL benchmarks for passage and document retrieval tasks.

**More Details on OpenQA Experiments:** All the DPR related experimental settings, baseline systems, and DPR Reader are based on their open source libarary[2]. The RAG-Token reader uses their open-source release in huggingface[3]. The RAG-Seq release in huggingface is not yet stable by the time we did our experiment, thus we choose the RAG-Token in our OpenQA experiment. RAG only releases the NQ models thus we use DPR reader on TriviaQA. We feed top 20 passages from ANCE to RAG-Token on NQ and top 100 passages to DPR's BERT Reader, following the guideline in their open-source codes.

**More Details on Baselines:** The most representative sparse retrieval baselines in TREC DL include the standard BM25 ("bm25base" or "bm25base_p"), Best TREC Sparse Retrieval ("bm25tuned_rm3" or "bm25tuned_prf_p") with tuned query expansion (Lavrenko & Croft, 2017), and Best DeepCT ("dct_tp_bm25e2", doc only), which uses BERT to estimate the term importance for BM25 (Dai & Callan, 2019a). These three runs represent the standard sparse retrieval, best classical sparse retrieval, and the recent progress of using BERT to improve sparse retrieval. We also include the standard cascade retrieval-and-reranking systems BERT Reranker ("bm25exp_marcomb" or "p_exp_rm3_bert"), which is the best run using standard BERT on top of query/doc expansion, from the groups with multiple top MARCO runs (Nogueira & Cho, 2019; Nogueira et al., 2019).

**BERT-Siamese Configurations:** We follow the network configurations in Luan et al. (2020) in all Dense Retrieval methods, which we found provides the most stable results. More specifically, we initialize the BERT-Siamese model with RoBERTa base (Liu et al., 2019) and add a $768 \times 768$ projection layer on top of the last layer's "[CLS]" token, followed by a layer norm.

We use BERT-Siamese, NLL loss, and dot product to be consistent with recent research. We have obtained better accuracy with more vectors per document, BCE loss, and cosine similarity, but that is not the focus of this paper.

**Implementation Details:** The training often takes about 1-2 hours per ANCE epoch, which is whenever new ANCE negative is ready, it immediately replaces existing negatives in training, without waiting. It converges in about 10 epochs, similar to other DR baselines. The optimization uses LAMB optimizer, learning rate 5e-6 for document and 1e-6 for passage retrieval, and linear warm-up and decay after 5000 steps. More detailed hyperparameter settings can be found in our code release.

## A.2 OVERLAP WITH SPARSE RETRIEVAL IN TREC 2019 DL TRACK

As a nature of TREC-style pooling evaluation, only those ranked in the top 10 by the 2019 TREC participating systems were labeled. As a result, documents not in the pool and thus not labeled are all considered irrelevant, even though there may be relevant ones among them. When reusing TREC style relevance labels, it is very important to keep track of the "hole rate" on the evaluated systems, i.e., the fraction of the top K ranked results without TREC labels (not in the pool). A larger hole rate shows that the evaluated methods are very different

---

[2]https://github.com/facebookresearch/DPR.
[3]https://huggingface.co/transformers/master/model$_doc/rag.html$

Table 6: Coverage of TREC 2019 DL Track labels on Dense Retrieval methods. Overlap with BM25 is calculated on top 100 retrieved documents.

| | TREC DL Passage | | | TREC DL Document | | |
|---|---|---|---|---|---|---|
| **Method** | **Recall@1K** | **Hole@10** | **Overlap w. BM25** | **Recall@100** | **Hole@10** | **Overlap w. BM25** |
| BM25 | 0.685 | 5.9% | 100% | 0.387 | 0.2% | 100% |
| BM25 Neg | 0.569 | 25.8% | 11.9% | 0.217 | 28.1% | 17.9% |
| BM25 + Rand Neg | 0.662 | 20.2% | 16.4% | 0.240 | 21.4% | 21.0% |
| ANCE (FirstP) | 0.661 | 14.8% | 17.4% | 0.266 | 13.3% | 24.4% |
| ANCE (MaxP) | - | - | - | 0.286 | 11.9% | 24.9% |

Table 7: Results of different hyperparameter configurations. "Top K Neg" lists the top k dense retrieved candidates from which we sampled the ANCE negatives from.

| | Hyperparameter | | | MARCO Dev Passage Retrieval MRR@10 | TREC DL Document Retrieval NDCG@10 |
|---|---|---|---|---|---|
| | **Learning rate** | **Top K Neg** | **Refresh (step)** | | |
| **Passage ANCE** | 1e-6 | 200 | 10k | **0.33** | – |
| | 1e-6 | 500 | 10k | 0.31 | – |
| | 2e-6 | 200 | 10k | 0.29 | – |
| | 2e-7 | 500 | 20k | 0.303 | – |
| | 2e-7 | 1000 | 20k | 0.302 | – |
| **Document ANCE** | 1e-5 | 100 | 10k | – | 0.58 |
| | 1e-6 | 100 | 20k | – | 0.59 |
| | 1e-6 | 100 | 5k | – | 0.60 |
| | 5e-6 | 200 | 10k | – | **0.614** |
| | 1e-6 | 200 | 10k | – | 0.61 |

from those systems that participated in the Track and contributed to the pool, thus the evaluation results are not perfect. Note that the hole rate does not necessarily reflect the accuracy of the system, only the difference of it.

In TREC 2019 Deep Learning Track, all the participating systems are based on sparse retrieval. Dense retrieval methods often differ considerably from sparse retrievals and in general will retrieve many new documents. This is confirmed in Table 6. All DR methods have very low overlap with the official BM25 in their top 100 retrieved documents. At most, only 25% of documents retrieved by DR are also retrieved by BM25. This makes the hole rate quite high and the recall metric not very informative. It also suggests that DR methods might benefit more in this year's TREC 2020 Deep Learning Track if participants are contributing DR based systems.

The MS MARCO ranking labels were not constructed based on pooling the sparse retrieval results. They were from Bing (Bajaj et al., 2016), which uses many signals beyond term overlap. This makes the recall metric in MS MARCO more robust as it reflects how a single model can recover a complex online system.

## A.3 HYPERPARAMETER STUDIES

We show the results of some hyperparameter configurations in Table 7. The cost of training with BERT makes it difficult to conduct a lot of hyperparameter explorations. Often a failed configuration leads to divergence early in training. We barely explore other configurations due to the time-consuming nature of working with pretrained language models. Our DR model architecture is kept consistent with recent parallel work and the learning configurations in Table 7 are about all the explorations we did. Most of the hyperparameter choices are decided solely using the training loss curve and otherwise by the loss in the MARCO Dev set. We found the training loss, validation NDCG, and testing performance align well in our (limited) hyperparameter explorations.

## A.4 CASE STUDIES

In this section, we show Win/Loss case studies between ANCE and BM25. Among the 43 TREC 2019 DL Track evaluation queries in the document task, ANCE outperforms BM25 on 29 queries, loses on 13 queries, and ties on the rest 1 query. The winning examples are shown in Table 8 and the losing ones are in Table 9. Their corresponding ANCE-learned (FirstP) representations are illustrated by t-SNE in Fig. 6 and Fig. 7.

In general, we found ANCE better captures the semantics in the documents and their relevance to the query. The winning cases show the intrinsic limitations of sparse retrieval. For example, BM25 exact matches the "most popular food" in the query "what is the most popular food in Switzerland" but the document is about Mexico. The term "Switzerland" does match with the document but it is in the related question section.

The losing cases in Table 9 are also quite interesting. Many times we found that it is not that DR fails completely and retrieves documents not related to the query's information needs at all, which was a big concern when we started research in DR. The errors ANCE made include retrieving documents that are related just not exactly

Table 8: Queries in the TREC 2019 DL Track Document Ranking Tasks where ANCE performs better than BM25. Snippets are manually extracted. The documents at the first disagreed ranking positions are shown. ANCE won on all of them. The NDCG@10 of ANCE and BM25 in the corresponding query is listed.

|  | ANCE | BM25 |
|---|---|---|
| **Query:** | qid (104861): Cost of interior concrete flooring | |
| Title: | Concrete network: Concrete Floor Cost | Pinterest: Types of Flooring |
| DocNo: | D293855 | D2692315 |
| Snippet: | For a concrete floor with a basic finish, you can expect to pay $2 to $12 per square foot... | Know About Hardwood Flooring And Its Types White Oak Floors Oak Flooring Laminate Flooring In Bathroom ... |
| Ranking Position: | 1 | 1 |
| TREC Label: | 3 (Very Relevant) | 0 (Irrelevant) |
| NDCG@10: | 0.86 | 0.15 |
| **Query:** | qid (833860): What is the most popular food in Switzerland | |
| Title: | Wikipedia: Swiss cuisine | Answers.com: Most popular traditional food dishes of Mexico |
| DocNo: | D1927155 | D3192888 |
| Snippet: | Swiss cuisine bears witness to many regional influences, ... Switzerland was historically a country of farmers, so traditional Swiss dishes tend not to be... | One of the most popular traditional Mexican deserts is a spongy cake ... (in the related questions section) What is the most popular food dish in Switzerland?... |
| Ranking Position: | 1 | 1 |
| TREC Label: | 3 (Very Relevant) | 0 (Irrelevant) |
| NDCG@10: | 0.90 | 0.14 |
| **Query:** | qid (1106007): Define visceral | |
| Title: | Vocabulary.com: Visceral | Quizlet.com: A&P EX3 autonomic 9-10 |
| DocNo: | D542828 | D830758 |
| Snippet: | When something's visceral, you feel it in your guts. A visceral feeling is intuitive — there might not be a rational explanation, but you feel that you know what's best... | Acetylcholine A neurotransmitter liberated by many peripheral nervous system neurons and some central nervous system neurons... |
| Ranking Position: | 1 | 1 |
| TREC Label: | 3 (Very Relevant) | 0 (Irrelevant) |
| NDCG@10: | 0.80 | 0.14 |

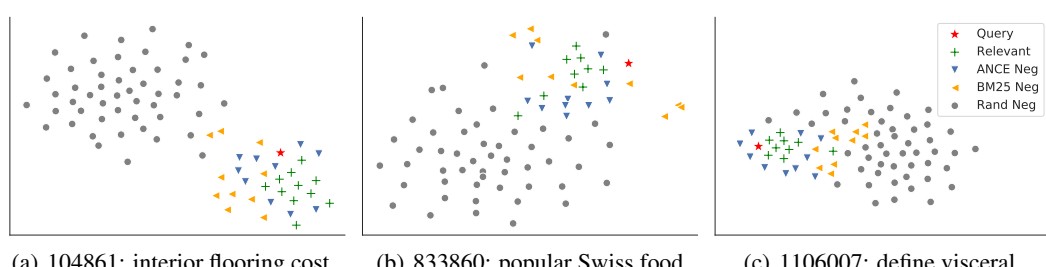

(a) 104861: interior flooring cost.  (b) 833860: popular Swiss food  (c) 1106007: define visceral

Figure 6: t-SNE Plots for Winning Cases in Table 8. Qids and queries are listed in the sub-captions.

relevant to the query, for example, "yoga pose" for "bow in yoga". In other cases, ANCE retrieved wrong documents due to the lack of the domain knowledge: the pretrained language model may not know "active margin" is a geographical terminology, not a financial one (which we did not know neither and took some time to figure out when conducting this case study). There are also some cases where the dense retrieved documents make sense to us but were labeled irrelevant.

The t-SNE plots in Fig. 6 and Fig. 7 show many interesting patterns of the learned representation space. The ANCE winning cases often correspond to clear separations of different document groups. For losing cases the representation space is more mixed, or there is too few relevant documents which may cause the variances in

Table 9: Queries in the TREC 2019 DL Track Document Ranking Tasks where ANCE performs worse than BM25. Snippets are manually extracted. The documents at the first positions where BM25 wins are shown. The NDCG@10 of ANCE and BM25 on the corresponding query is listed. Typos in the query are from the realist web search queries in TREC.

| | ANCE | BM25 |
|---|---|---|
| **Query:** | qid (182539): Example of monotonic function | |
| Title: | Wikipedia: Monotonic function | Explain Extended: Things SQL needs: sargability of monotonic functions |
| DocNo: | D510209 | D175960 |
| Snippet: | In mathematics, a monotonic function (or monotone function) is a function between ordered sets that preserves or reverses the given order... For example, if y=g(x) is strictly monotonic on the range [a,b] ... | I'm going to write a series of articles about the things SQL needs to work faster and more efficienly... |
| Ranking Position: | 1 | 1 |
| TREC Label: | 0 (Irrelevant) | 2 (Relevant) |
| NDCG@10: | 0.25 | 0.61 |
| **Query:** | qid (1117099): What is a active margin | |
| Title: | Wikipedia: Margin (finance) | Yahoo Answer: What is the difference between passive and active continental margins |
| DocNo: | D166625 | D2907204 |
| Snippet: | In finance, margin is collateral that the holder of a financial instrument ... | An active continental margin is found on the leading edge of the continent where ... |
| Ranking Position: | 2 | 2 |
| TREC Label: | 0 (Irrelevant) | 3 (Very Relevant) |
| NDCG@10: | 0.44 | 0.74 |
| **Query:** | qid (1132213): How long to hold bow in yoga | |
| Title: | Yahoo Answer: How long should you hold a yoga pose for | yogaoutlet.com: How to do bow pose in yoga |
| DocNo: | D3043610 | D3378723 |
| Snippet: | so i've been doing yoga for a few weeks now and already notice that my flexiablity has increased drastically. ...That depends on the posture itself ... | Bow Pose is an intermediate yoga backbend that deeply opens the chest and the front of the body...Hold for up to 30 seconds ... |
| Ranking Position: | 3 | 3 |
| TREC Label: | 0 (Irrelevant) | 3 (Very Relevant) |
| NDCG@10: | 0.66 | 0.74 |

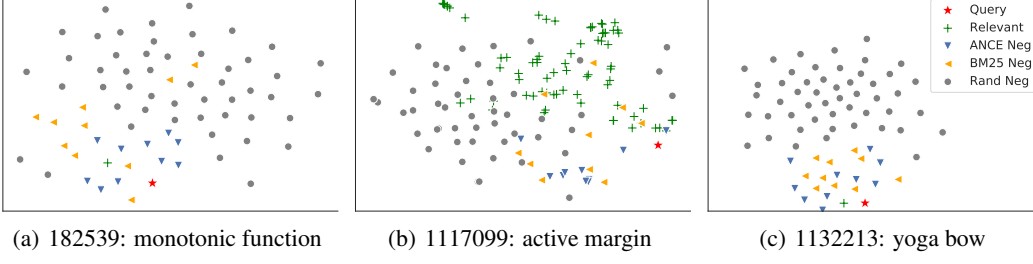

(a) 182539: monotonic function     (b) 1117099: active margin     (c) 1132213: yoga bow

Figure 7: t-SNE Plots for Losing Cases in Table 9. Qids and queries are listed in the sub-captions.

model performances. There are also many different interesting patterns in the ANCE-learned representation space. We include the t-SNE plots for all 43 TREC DL Track queries in our open-source repo. More future analyses of the learned patterns in the representation space may help provide more insights on dense retrieval.

