# OpenReview forum: "Approximate Nearest Neighbor Negative Contrastive Learning for Dense Text Retrieval"
_ICLR.cc/2021/Conference — ICLR 2021 Poster_

### Official Review · AnonReviewer1 · 2020-10-19
**ANCE for Text Retrieval**

**Rating:** 6
**Confidence:** 3

**Review:**

This paper studies the problem of representation learning for first-stage retrieval in text ranking/matching. Specifically, it investigates the role of negative instances and how to select them in the quality of representations. The paper is well-written and it does a fair job in motivating the problem and discussing related works.

The main pros are as follows:

* Authors attempt to theoretically study the importance of negative instances and their impact on the gradients of objective function for text retrieval. Specifically, they posit that non-informative negative instances provide small gradients n the course of training, and thus adversely affect the convergence of gradient-based learnings.

* Through various experiments, the authors show that by exploiting the top retrieved texts from the corpus, using the current learned representations, as 'hard' negative instances, the performance of text ranking methods can be significantly improved.

There are however some ambiguities and room for improvement:

* As equation (10) suggests, the optimal distribution over negative instances is the one that minimizes the variance of gradients, or equivalently the norm of loss function. However, at many places it is pointed out that small losses prevent the model from learning and slow down the convergence rate. This is a kind of contradiction, and needs to be clarified,

* Existing results from literature such as equation (11) are leveraged to connect the norm of loss function and the gradient of last layer of representation learning models. However, the arguments here are not quite rigorous. It should be clarified that these results hold for what types of ranking objectives, and what other assumptions such as smoothness, etc. are required.

* The main bottleneck of using top retrieved documents as negative instances is the computational burden of updating the ANN indexing per batch. Therefore, the authors propose to perform this updating less frequently. It would be nice to have more comprehensive experiment to show that how sensitive is the performance vs the frequency of index updates.

---

> ### Author Response · Authors · 2020-11-17
> **Author Responses to Reviewer 1: Improved Clarity, More Details in Theory Assumptions & Empirical Verification, and Additional Analyses on Index Update Frequency**
>
> Dear Reviewer 1:
>
> We have updated a new version in the system including additional experiments and revisions to improve clarity. The full list of updates is in the general comments. Note that we have added new results in end-to-end OpenQA showing that ANCE’s improved text retrieval can propagate to later Question Answering modules for better answer accuracy.
>
> Thanks for pointing out those ambiguities. We updated our papers to improve clarities on them.
>
> On the ambiguity between gradient estimation variance and gradient norm:
>
> * First, to clarify, we use the norm of per instance gradient norm, not “norm of loss function”. We revised Section 3.2 and added Eqn. 13 to sum up the logic of our reasoning chain. We also made it more clear on the direction of changes: smaller loss -> small gradient norm -> large scholastic variance of the gradient estimator -> slower convergence.
> * The oracle importance sampling is the minimize *the variance of the scholastic gradient estimator*, and equally sampling proportional to *per-instance gradient norm* (not norm of loss).
>   * The estimator refers to the estimation of the full data gradient (on all negatives) using the sampled negatives. Its variance is a statistical variable about the properties of mini-batch/scholastic training w.r.t. full batch training in non-convex optimization.
>   * The per-instance gradient norm is the L2 norm of the actual gradient on each data point. Sampling negatives with small loss and thus smaller per-instance gradient norms deviates from the oracle importance sampling, thus leads to sub-optimal learning convergence.
> * We add the description of “the variance of the scholastic gradient estimator” under Eqn. 10 to avoid confusing it with gradient norm.
>
>
> About the upper bound of the gradient norm using the norms of the last layer (note the Eqn. 11 in previous version is now Eqn. 12):
>
> * We added more details on the conditions used to derive the upper bound after Eqn. 12. The form derivation was done on multi-layer MLP with regular activation functions by Katharopoulos & Fleuret (2018). All our other theorical analyses, i.e. those in Eqn. 13, except this upper bound, are applicable on general neural networks following the importance sampling framework.
> * We agree that it will be ideal if this upper bound can be proved on complex neural networks such as deep transformers. More theorical analyses on deep transformers is sure a much-needed future research. Nonetheless, in the updated Fig. 4, we show that this upper bound empirically holds in BERT-Siamese in the updated Fig. 4.
>  * Note that we update the gradient norms of other transformer layers to the average of its network layers, including the feedforwards and Q/K/V, to have a fair comparison with the gradient on the last feedforward layer. In the last version the plotted gradient norms are the sum instead of their average.
>
>
> On the sensitivity of learning to the ANN index update frequency:
>
> * We moved our studies on the sensitivity to the index refresh rate from Appendix A.3 to the main paper, in Section 6.4. We found that, empirically, a desirable refresh rate can be achieved if we allocated equal number of GPUs to Trainer and Inferencer, such as four or eight on each. More frequent refresh does not improve learning convergence in our experiments.
> * We discuss this trade-off in Section 6.4. that using 4-8 extra GPUs for a one-time training cost is a good return of investment in many real search scenarios.
> * In our production environment, where the index corpus is way bigger than Wikipedia and MSMARCO, we can further reduce the cost of the index refresh by discarding those low-quality web pages or further sub-sample the corpus.   Those web pages, e.g., with low authority scores, are unlikely to be hard negatives for any query. This is one of many engineering tricks we can apply to improve the efficiency in large scale systems if needed.
>
> Thanks again for you review. Please let us know if you have any further suggestions.

---

### Official Review · AnonReviewer2 · 2020-10-26
**The work has values in better performance and open source of the proposed method, though it should justify further the performance gain. I would like to vote for a weak accept.**

**Rating:** 7
**Confidence:** 3

**Review:**

##########################################################################

Summary:

This paper studies the problem of dense text retrieval, which represents texts as dense vectors for approximate nearest neighbors (ANN) search. Dense text retrieval has two phases. The first phase learns a representation model to project semantically similar texts to vectors of large similarity scores (e.g. inner products or cosine similarity scores). The second phase adopts an ANN search algorithm to index these vectors and process queries. The paper claims key contributions at the first phase. Specifically, (1) The paper introduces a better negative sampling method to sample good dissimilar text pairs for training.  (2) The new method enables faster converge of model learning. (3) The new method leads to 100x faster efficiency than a BERT-based baseline, while achieving almost the same accuracy as the baseline.

##########################################################################

Reasons for score:

Overall, I like the idea of this paper and opt for a weak accept. A carefully designed negative sampling method should be able to outperform baselines that use simple heuristics. The efficiency improvement 100X is very promising. However, the paper can be better in experimental comparison and presentation. For experimental comparison, a stronger baseline using dense vectors should be included to strengthen the performance claim. For presentation, many important terms require clear definitions, without which the performance gain is not understandable. It will be good if the authors can address the above two issues in the rebuttal.

##########################################################################

Pros:

1. The paper proposes a novel negative sampling method. Based on the method, the paper proposes a new dense text retrieval framework ANCE. ANCE introduces an asynchronous index refresh to select the most dissimilar text pairs for training in a timely manner.

2. The proposed ANCE achieves faster model training and equally accurate text retrieval when compared with a number of baselines. In a TREC 2019 task, ANCE achieves the best NDCG score against 11 baselines.

3. The authors promise to make code open source. That will greatly improve the reproducibility of the work.  The code, together with its performance, will serve as a new state-of-the-art for future study.

##########################################################################

Cons:

1.	An important baseline is missing. In section 5, the paper describes Baselines. According to the descriptions, all baseline use BM25 to retrieve samples for training. BM25 may not be the best for a strong baseline since it relies on sparse word tokens. An alternative is to use BERT [CLS] dense vectors of all texts and a similarity search algorithm such as locality sensitive hashing as the retriever. It will be good if the authors can add the baseline to the paper.

2. The paper does not explain clearly why the proposed method runs faster than baselines. The experimental results support that the proposed method outperforms several baselines. However, the paper does not explain the performance superiority. I am not sure about which of dissimilar text pairs selection or index refresher or others in the proposed negative sampling leads to the superiority. The reason for uncertainty may be due to the lacking of definitions in the paper. For example, “BERT rerank” refers to a baseline but it has not a definition in the paper. The input and output of “BERT rerank” remain unclear. Similarly, “TREC 2019” is an important benchmark but it has not definitions related to the inputs and outputs. It is necessary to explain important concepts for the best readability of the paper.

##########################################################################

Questions during rebuttal:

I would like to see some experiments or discussions to clarify the above cons.

---

> ### Author Response · Authors · 2020-11-17
> **Author Response to Reviewer 2: Clarification on Dense Retrieval Baselines and More Discussions on the Efficiency Advantage of Dense Retrieval**
>
> Dear Reviewer 2: We updated a new version with additional experiments and improved clarity. The full list of updates is in the general comments. We added more details and discussions on the used DR architecture and its difference with BERT Rerankers to avoid confusion. Thanks a lot for pointing them out.
>
>
>
> On the dense retrieval baseline:
> * All the dense retrieval baselines as well as ANCE (FirstP) use exactly the BERT similarity model you suggested.
> * Only the DPR+BM25 baseline in OpenQA retrieval is a fusion of BM25 with DPR results, released by the DPR authors. The fusion is a standard way to combine dense retrieval with sparse retrieval and often leads to better results as the two captures quite different signals (as studied in Appendix A.2). We do not fusion ANCE with any sparse retrieval or any other retrieval systems in our experiments for more direct comparisons.
> * In other DR systems, BM25 is only used to provide negatives in training.
> * We have made various updates in the paper to improve the clarity on this front:
>  * We add a dedicated passage in Section 2 to describe and define BERT-Siamese.
>  * We refer to the BERT-Siamese definition (Eqn. 2) when describing the DR baselines in Section 5.
>
>
> On the difference with BERT reranker and the efficiency improvements:
> * BERT reranker applies BERT on the concatenation of a query-document pair, e.g., for each candidate document, BERT reranker scores it using BERT(q \cat d). This makes it more suitable in the reranking stage, which is to score the top retrieved candidates (100 in our experiments) from first stage retrieval.
> * Even in reranking stage, the BERT reranker has to run inference on each of the candidate documents concatenated with the query, which is 100 BERT inference operation if reranking top 100 candidate documents. The inference of each document is query-dependent and must be done online, unless we cache the q-d’s ranking score.
> * The efficiency gain we measured is in the online phrase, when the models are used to serve in a search engine . ANCE, or BERT-Siamese in general, has two intrinsic advantages in efficiency over BERT reranker:
>  * The encoding of documents can be computed offline as they are query independent. In serving time, only query encodings and the dot prod ANN search are required.
>  * The ranking score of different q-d pairs in BERT-Siamese use the same query and document encoding, while BERT-rerank needs to rerun BERT on each pair.
> * The paper updates improve the clarity on this front in various places:
>  * We add the definition of BERT reranker in Sec. 2, after BERT-Siamese (Eqn. 2).
>  * We add a more detailed discussion on the efficiency comparison in a dedicated subsection: Section 6.2.
>
>
> In addition, we revise the following part of the paper to resolve the ambiguity you pointed out:
> * We add more description of the index refreshing implementation in the last paragraph of Section 5.
> * We add discussion on the trade-off in asynchronous training and its analyses in Section 6.4.
> * We add more description of the web track in the Benchmark paragraph of Section 5.
>
>
>
> Thanks again for your review. Please let us know if you have further questions and suggestions.

---

### Official Review · AnonReviewer4 · 2020-10-29
**Mining Hard Negatives improves retrieval performance**

**Rating:** 9
**Confidence:** 5

**Review:**

##########################################################################
Summary:
Authors start from an assumption: “local negative sampling is the bottleneck of dense retrieval’s effectiveness”. To overcome this limitation, authors propose ANCE (Approximate nearest neighbour Negative Contrastive Estimation), a new contrastive representation learning mechanism for dense retrieval. The basic idea is that of constructing negatives exploiting the being trained deep retrieval module. The idea is that the model considers as negatives borderline cases. They also show, theoretically, that this improves the variance of the stochastic gradient estimation thus leading to faster convergence.

##########################################################################
Reasons for score:
I honestly enjoyed reading the paper. It has a theoretical justification that explains the intuition of using hard negatives. Experiments are thorough and they show improvements over the state of the art. The discussion section is thorough. I believe that this research results are very important also in practice


##########################################################################
Pros:

1. The paper addresses a timely and important problem
2. The paper gives a nice theoretical justification for the reasons why they have to use hard negatives
3. Experiments are thorough and nicely done. Results are very good. I particularly enjoyed seeing that both on public datasets and on real world search systems the novel retrieval mechanism helps greatly.

##########################################################################
Cons:

1. The only one thing that I believe might impair the utilisation of this method is that you need to reconstruct the embeddings every m batches. It takes 10h every reconstruction and it is not clear what happens every m batches. Do you start a novel reconstruction? Do you replace the current embedding version with a new one as soon as one finishes training? This aspect, in my opinion, is the weakest of the paper and it would deserve more attention by the authors.

##########################################################################
Questions during rebuttal period:

Please address and clarify the cons above

#########################################################################
Some minor issues
(1) In equation (2) I would call D^+ and D^-, D_q^+ and D_q^- in order to remark that negatives and positives are per-query.
(2) an negative —> a negative
(3) Citation Luan et al. —> It’s Toutanova not Toutanove

---

> ### Author Response · Authors · 2020-11-17
> **Author Response to Reviewer 4: Updated Paper with More Implementation Details and Studies on the Asynchronous Training**
>
> Dear Reviewer 4:
>
> We have updated a new version with added experiments, analyses, and revised writing. The detailed change list is in the general comment. Those most related to your questions are update #3 and #10 on the analyses of embedding reconstruction and its implementation details. We have fixed the typos and make the notation clearer following your suggestions.
>
> More discussions to your question:
> * The reconstruction of the embedding in the ANCE negative index is also paralleled on multiple GPUs. The forward pass is easy to parallel and in our experiments, we often use four or eight GPUs, equal to those used in the training side in this reconstruction. This reduces the reconstruction time to 2.5 or 1.25 hours. (We measure the time on one GPU in Table 5 for fair comparisons.)
> * Each reconstruction uses a new model checkpoint to inference. No encodings of the document from the last model checkpoint are reused or cached. This is a place further optimization can be done. In the scale of Wikipedia, we found it efficient enough.
> * In production the corpus is much bigger, e.g. the entire web corpus of a commercial search engine. We can subsample the corpus and only keep the high-quality ones as another trade-off dimension. Some of our production experiments use this trick. We found a filtered corpus at the scale of ~100 million documents shows a good balance of efficiency and effectiveness in training.
> * As soon as the document encodings are reconstructed and the ANN index is built, the Inferencer writes the ANN index to a shared file system. The Trainer detects this and switches to the new index almost instantly (delayed only by how frequent the Trainer checks the ANN index updates). The communication between the two are infrequent and can be done through a shared file system. It is easy to parallelize them on multiple machines.
> * We moved the study on the impact of this index refreshing rate from Appendix A.3 to Section 6.4. We found that an equal allocation of GPUs to the index refreshing leads to a small enough async-gap that does not influence the training convergence much. We experimented with more frequent index refreshing and did not observe further gains.
> * We add some discussion on the trade-off in using one extra set of GPUs for index refreshing in Section 6.4. It is a one-time training cost that leads to significantly improved retrieval accuracy across the board. The DR models are also fine-tuned from pretrained models, which is way cheaper than pretraining. The bottleneck in dense retrieval systems is more on accuracy and serving than training efficiency.
>
> Thanks again for your review and please let us know if you have any further suggestions.

---

### Official Review · AnonReviewer3 · 2020-10-29
**Strong empirical results but idea seems incremental**

**Rating:** 6
**Confidence:** 4

**Review:**

The paper explores how to effectively do negative sampling for dense retrieval. The paper shows that negatives sampled locally in batch are not informative, and proposes ANCE, a learning mechanism that selects hard training negatives globally from the entire corpus, using an asynchronously updated ANN index.

Strength:
1) Experiments show that the proposed method significantly outperforms state-of-the-art approaches such as DPR on MSMARCO, TRECDL, NaturalLanguages and TriviaQA.

Weakness:
1) The analysis on why in-batch local negatives are ineffective (in section 3) does not seem to be very insightful. Also, I did not see a big connection between this analysis and the negative sampling technique proposed in the following section.
2) The idea of maintaining a set of global negatives is not new, and refreshing index asynchronously has also been explored in [Guu et al. 2020]

Overall, I think this is a borderline paper. The experiments show improved performance over baseline methods, but the idea seems a bit incremental (a combination of existing tricks for training retrieval model).

---

> ### Author Response · Authors · 2020-11-17
> **Author Response to Reviewer 3: Improved Clarity and More Discussions on the Contribution of ANCE**
>
> Dear Reviewer 3:
>
> We have updated a new version of this draft with new experiments and improved writing. The list of updates is in the general comments.
>
> About the analysis on the ineffectiveness of local negatives:
> * We have revised Section 3 to add more details and discussion on the theoretical analyses.
> * We added Eqn. 13 to recap our insights, and use it to motivate the needs of constructing harder, global negatives.
>
> We do not claim the novelty in refreshing the ANN index asynchronously. It is a technique that used both by us and REALM, but for different purposes.
> * ANCE focuses on representation learning for dense retrieval and uses the ANN index to construct global hard negatives for contrastive learning. REALM focuses on grounded language modeling and uses the ANN index to find grounding documents.
> * The retrieval part in REALM is indirectly trained by the language model loss, while ANCE focus on the retrieval itself. The comparison in the End-to-end QA accuracy shows that ANCE provides better retrieval accuracy which leads to better answer accuracy than REALM.
>
> On the contribution of this work:
> * It is intuitive that improving the difficulty of negatives is useful. Nevertheless, we are among the first to theoretically show the needs of harder negatives and the intrinsic limitation of the widely used local negatives in contrastive learning.
> * To the best of our knowledge we are among the first to provide both theorical analyses and an effective solution to construct global negatives in contrastive representation learning.
>
> There are also several few parallel work that study how to construct harder negatives in contrastive learning:
> * Kalantidis et al. [1] is posted on ArXiv the same day as ICLR 2021 submission deadline and is to appear in NeurIPS in next month. They argue that, in the abstract: “an important aspect of contrastive learning, i.e. the effect of hard negatives, has so far been neglected”. They followed their argument and developed a new technique to construct synthetic negatives by mixing the top negatives and positives in the vision representation feature space.
>   * Very interestingly, their data analyses in Fig. 2a show that the ImageNet dataset has some common properties with text retrieval (our Fig. 3), showing the needs of global negatives exist in both text and computer vision tasks, both have been neglected in recent research.
>   * Our submission shares a similar motivation but provides a different solution that directly select globally hard negatives.
>   * Our theoretical analyses also back up their motivation.
>   * Our focus is on text data where the vision-style feature level mixing is not as straightforward, e.g., mixing two negative documents in a brute-force way is unlikely to create harder negatives.
> * There is another parallel submission to ICLR 2021 [2] that also studies how to sample informative negatives in contrastive learning.
>   * They provide theoretical analyses from a different angel, mainly in the image representation learning task.
>   * They develop a solution quite different from ours. It reweights the local negative samples based on their hardness. In comparison, we directly sample negatives from those globally hardest ones.
> Those parallel papers show that how to construct better negative training instances is a crucial and on-going research topic in representation learning. Our submission provides a new angle from the importance sample perspective, which is also empirically verified in our experiments. We also develop a new simple solution to construct global negatives using a different trade-off (async-gap versus synthetic, e.g..), which show strong performance in various academic benchmarks and a real production environment.
>
>
> [1] Yannis Kalantidis, Mert Bulent Sariyildiz, Noe Pion, Philippe Weinzaepfel, and Diane Larlus. Hard negative Mixing for Contrastive Learning. To appear in NeurIPS 2020. Posted on ArXiv Oct 2nd, 2020.
>
> [2] Anonymous submission to ICLR 2021. Contrastive Learning with Hard Negative Samples. https://openreview.net/forum?id=CR1XOQ0UTh-

---

### Author Response · Authors · 2020-11-17
**New Experiments and Writing Updates in the Revised Version**

Thanks for your feedback. We have updated our submission with new experimental results, additional analyses, and improved writing clarity. The changes are summarized as follows.

New experiments and updated analyses:

1.	We show ANCE’s better retrieval leads to better end-to-end OpenQA accuracy in Table 4. We evaluated ANCE’s effectiveness in the end-to-end OpenQA accuracy on Natural Questions (NQ) and Trivia QA (TQA). We swap the DPR retriever in RAG and DPR systems to ANCE retriever, kept the reading component the same (which is RAG-Token reader on NQ and DPR reader on TQA), and achieved ~1 point QA Exact Match gains on both. The end-to-end results on NQ also provides an empirical reference to REALM, whose asynchronous index refresh is used to support ANCE, but focuses on grounded language modeling not contrastive representation learning.
2.	We empirically confirm the theoretical upper bound in Eqn. 12 with updated Figure 4. We plot the gradient norm of BERT-Siamese last layer, the MLP projection, when training with ANCE (FirstP). We also update the gradient norm of the other transformer layer groups to the average of per layer (e.g., per feed-forward or Q/K/V layer in the hybrid transformer layer) for a direct comparison. It shows that the gradient norm of the last layer in BERT-Siamese is indeed much higher than the lower layers, which empirically aligns with the theoretical upper bound in Eqn. 12.
3.	We move the study on ANCE’s asynchronous training from Appendix to Section 6.4. We show that with an equal (four or eight on each) GPU allocation in the training side (Trainer) and the ANCE index refreshing side (Inferencer) and suitable learning rate, the learning convergence is not sensitive to the asynchronous gap (async-gap) between the negative index and the being optimized DR model. Further reducing the async-gap does not improve training convergence.


Writing updates to improve clarity:


4.	We add a paragraph in Section 2, “BERT-Siamese Model”, to describe and define the dense retrieval (DR) model architecture used in the DR baselines and ANCE. All the DR baselines in web search use this exact same model and we refer to Eqn. 2 in various places to avoid confusion.
5.	We split Section 3 into two subsections to make the theoretical analyses more streamlined.
6.	We add more details about the upper bound of gradient norm (Eqn. 12), including the conditions used to derive it, as well as the empirical verification on BERT-Siamese in Section 6.3.
7.	We revise our wording throughout Section 3 to distinguish “per-instance gradient norm” and “the variance of the scholastic variance estimator”. The former is the norm of the actual gradient on one data point in mini-batch training. The latter is the variance of the scholastic/sampling estimator of the gradient on full data, which categorizes the property of the negative sampling in estimating the full non-scholastic gradient.
8.	We add Eqn. (13) to recap our theoretical analyses. We added the direction of variable changes to improve clarity.
9.	We add more description on the web search task in the Benchmark paragraph of Section 5.
10.	We add the last paragraph in Section 5 to describe the implementation details of the ANN index refreshing and the communication between the Trainer and the Inferencer during ANCE’s asynchronous training.
11.	We split the efficiency evaluation to an individual Section 6.2. We add more discussions on the BERT-ranker model in both Section 2 and Section 6.2, with more details about its algorithmic complexity and its cost in online serving.
12.	We fixed the typos and updated some notations following reviewers’ suggestions. We conducted several additional arounds of proofread to improve writing quality.

---

### Decision · Program_Chairs · 2021-01-07
**Final Decision**

**Decision:**

Accept (Poster)

**Comment:**

The paper explores how to effectively conduct negative sampling in learning for text retrieval. The paper shows that negative examples sampled locally are not informative, and proposes ANCE, a new learning mechanism that samples hard negative examples globally, using an asynchronously updated ANN index.

Pros • The problem studied is important. • Paper is generally clearly written. • Solid experimental results. • There is theoretical analysis.

Cons • The idea might not be so new. The contribution is mainly from its empirical part.

During rebuttal, the authors have addressed the clarity issues pointed out by the reviewers. They have also added additional experimental results.